# Sugar-Sweetened Beverages Consumption and Associated Factors among Northeastern Chinese Children

**DOI:** 10.3390/nu13072233

**Published:** 2021-06-29

**Authors:** Xuxiu Zhuang, Yang Liu, Joel Gittelsohn, Emma Lewis, Shenzhi Song, Yanan Ma, Deliang Wen

**Affiliations:** 1Institute of Health Science, China Medical University, Shenyang 110000, China; cxzhuang@cmu.edu.cn (X.Z.); yliu0568@cmu.edu.cn (Y.L.); szsong@cmu.edu.cn (S.S.); ynma@cmu.edu.cn (Y.M.); 2Human Nutrition Center, Department of International Health, Johns Hopkins Bloomberg School of Public Health, Baltimore, MD 21205, USA; jgittel1@jhu.edu (J.G.); elewis40@jhmi.edu (E.L.)

**Keywords:** sugar-sweetened beverages, home-related factors, community environment, Northeastern Chinese children

## Abstract

(1) Background: The present study aimed to investigate the association between home-related factors, community environmental factors, and sugar-sweetened beverages (SSBs) intake among Northeastern Chinese children. (2) Methods: Cross-sectional. Children with complete data were included in the analysis (*n* = 901). A questionnaire modified according to BEVQ-15 measured the intake of SSBs. Logistic regression was applied to determine the factors associated with the consumption of SSBs. IBM SPSS Statistics 23.0 was applied to perform all statistical analyses. (3) Results: The mean total amount of SSBs consumed on a weekly basis was 2214.04 ± 2188.62 mL. Children’s weekly pocket money, frequency of SSBs purchase, SSBs availability at home, the number of accessible supermarkets, and frequency of weekly visits to convenience stores were all found to be associated with a high intake of SSBs among all children. Among children of normal weight, the findings indicated that weekly pocket money, SSBs availability at home, and number of accessible supermarkets were associated with a high SSBs intake. At the same time, frequency of SSBs purchase, mother’s SSBs intake, and frequency of weekly visits to convenience stores were associated with a high SSBs intake among children with obesity. (4) Conclusions: Given the potential negative health effects of high SSBs intake, it is crucial to pay attention to home-related factors and community environment.

## 1. Introduction

The childhood obesity epidemic, considered as a major public health challenge of the 21st century, has become one of the most serious health-related problems in many countries [1,2]. With the rapid development of China’s economy in recent years, the prevalence of overweight and obesity among Chinese children has become increasingly serious. In 2015, China was deemed to be the country with the largest number of obese children in the world [3].

High consumption of sugar-sweetened beverages (SSBs) among adolescents has turned into a global concern due to its negative impact on health. Some studies show that higher intake of SSBs may increase risk of weight gain and obesity. [4]. According to the Food Guide Pagoda for Chinese Residents (2016), SSBs refer to beverages with added artificial sugar or beverages with sugar content of more than 5% added in the process [5]. Examples of SSBs include tea drinks, carbonated drinks, sweetened milk tea, milk drinks, coffees, sweetened fruit juices, sodas, sport drinks, energy drinks, fruit-flavored drinks, and beverages with added sugar [6,7,8,9]. Apart from its impact on obesity, high consumption of SSBs among children may lead to increased risk of insulin resistance, dental caries, sleep disturbance, and caffeine-related health concerns [10,11,12,13,14].

Importantly, however, much of the evidence for the adverse impact of SSBs on children comes from western countries, while evidence for the adverse impact of SSBs on children in China is limited. Evidence that does exist in China primarily focus on the consumption of only a few specific drinks, ignoring other types of SSBs [15,16,17].

Evidence from countries around the world shows that factors such as physical activity level [18,19,20,21], sleep quality [22,23], fast food intake [24,25], availability and accessibility of SSBs at home [25,26], parental role models [27,28], and peer pressure [28,29] are associated with SSB consumption among children. Nevertheless, only several studies have investigated the factors associated with SSBs intake among children in China. Due to differences in social culture and food culture between China and other countries, it is necessary to study the factors affecting SSBs intake in China. Home-related factors may be a particular influence on SSBs consumption among children [30]. The results of an analysis from the National Health and Nutrition Examination Survey (1999–2004) showed that 55% to 70% of all SSBs calories ingested by U.S. youth were consumed in the home environment, compared with 7% to 15% in schools [30]. Nevertheless, more studies are needed to better understand how factors such as the home environment are associated with SSB consumption.

The present study aimed to: (i) understand the consumption of SSBs among Northeastern Chinese children, and (ii) investigate the association between home-related factors, community environmental factors, and SSB intake among Northeastern Chinese children.

## 2. Materials and Methods

### 2.1. Study Population and Sampling

The cross-sectional study took place in two similar middle schools of Huanggu District in Shenyang city, China. The students were selected on the basis of the following criteria: (1) voluntarily participate in the study and sign the informed consent; (2) Secondary school students in grades 7–9 and their parents who can cooperate to ensure the completion of the questionnaire; (3) No plan to move out or transfer school in the near future (more than half a year). Exclusion criteria: (1) secondary obesity caused by congenital genetic diseases, metabolic diseases, endocrine diseases, etc.; (2) Patients with severe cardiovascular diseases, severe neurological diseases, and severe liver and kidney diseases; (3) Do not sign the informed consent and cannot cooperate with the personnel who are not willing to cooperate with the questionnaire survey; (4) Unable to cooperate with the investigators of this project due to mental factors; (5) Have recent plan to move out or transfer to another school (more than half a year).

Data were collected in July 2018 from 1020 school-aged adolescents (12–15 years old). In each school, only 88.3% of the expected participants responded due to absences or parental refusal. Adolescents with missing data (no height, no weight, no SSBs, inconsistent or no response: *n* = 119) were then excluded from this study. The final total sample was 901 adolescents.

All participants gave informed written consent prior to the participation in the study, in line with the legal requirements and the Declaration of Helsinki. The protocol was also approved by the Shenyang City Board of Education and the Ethics Committee of China Medical University.

### 2.2. Measurements and Variable Definitions

A set of questionnaires was completed by children and their parents to collect information with respect to sociodemographic background, home-related factors, community environmental factors, and consumption of SSBs. Household questionnaires about family demographics were answered by parents of students. An adapted version of the SSB questionnaire was answered by students. If an unqualified questionnaire was found, the parents will be given a gentle reminder. Those who still fail to meet the requirements will be treated as non-response. Written informed consent was obtained from all participants.

#### 2.2.1. Intake of Sugar-Sweetened Beverages

Students reported the amount and frequency of sugar-sweetened beverages intake on a weekly basis (for the past one-week). The questionnaire was developed using a modified version of the Beverages Intake Questionnaire (BEVQ-15). The BEVQ-15 estimates daily grams and energy intake of 15 beverages, including water and other sugar-sweetened beverages among adults [31]. The design of a newly modified version of the BEVQ-15 allowed for the measurements of sugar-sweetened beverages among Chinese children. The modified tool included 13 beverage categories (sweet fruit juice, tea drinks, coffee drinks, instant coffee, sweet milk tea, milk drinks, plant protein drinks, Asian specialty drinks, carbonated drinks, 100% fruit juice drinks, vegetable juices, low-calorie carbonated drinks, energy/sports drinks) and water. The Cronbach α was 0.901, and the internal consistency of the questionnaire was determined to be good. The half-reliability of the questionnaire was 0.786, indicating a general stability of the questionnaire.

To assess beverage consumption (“How much per week”), the modified BEVQ-15 used in present study asked, “For each beverage, please estimate how many bottles you drink based on the amount in the picture.” Children were asked to respond to: “How many bottles do you drink from Monday to Friday” and “How many bottles do you drink during Saturday and Sunday.” To determine frequency of SSBs intake (“How often”), the questionnaire asked, “In the past one-week, please estimate how often you drank the following beverages.” Response options were: “Never,” “1 to 2 times per week,” “3 to 4 times per week,” “5 to 6 times per week,” or “more than seven times per week.”

The average weekly total intake of SSB was calculated by multiplying the number of bottles per week by the amount per bottle of each beverage. Total beverage consumption was quantified by summing all beverages. The total intake of beverages is a skewed distribution, and the median was used to classify the total intake of SSBs.

#### 2.2.2. Home-Related Factors

In the present study, home-related factors refer to household income, pocket money, home availability of SSBs, and parental role modeling. Household income is a categorical variable, divided into three categories—(i) below 30,000 RMB, (ii) 30,000–100,000 RMB, and (iii) above 100,000 RMB. Home availability of SSBs was assessed by answering two questions, “How often parents purchased SSBs” and “Is the beverage placement easy to obtain?” Parental Role Modeling was measured using parental Saturday and Sunday SSB intake.

#### 2.2.3. Community Environmental Factors

In the present study, community environmental factors were assessed by using the following question: “Please recall the facilities in an area about 15 min (0.75 to 1.25 km) from your home and provide the number, walk time, frequency of visiting and the average cost per visit of fast food restaurant, supermarkets and convenience stores.” Responses were completed by parents.

#### 2.2.4. Child Overweight and Obesity

Body mass index (BMI) is calculated according to height and weight of children, BMI = weight (kg)/[height(m)]^2^. According to the Chinese overweight and obesity criteria recommended by the Chinese working group on obesity of International Life Sciences Institute, students were categorized as being either “normal weight,” “overweight,” or “obese” according to their age and gender. This is the unified standard for assessing overweight and obesity given BMI that has been used in China [32]. In the present study, children ages 10 to 15 were selected for investigation (Table 1).

### 2.3. Statistical Analysis

Participants with complete data were included in the analysis (*n* = 901). Individuals excluded from the analysis were not significantly different from those included. All statistical analyses were performed by the statistical software IBM SPSS Statistics Version 23 [33]. Continuous variables were presented as mean and standard deviation. Categorical variables are represented by count (*n*) and proportion (%). The associations between home-related factors, community environmental factors, and high SSBs intake were estimated by logistic models. The continuous variables were converted into categorical variables according to the median. Home-related factors were included in model 1, and home-related factors and community environmental factors were included in model 2. Age, sex, and total amount of water consumption were included as covariables in the model. Statistical significance was defined by a *p*-values threshold of <0.05.

## 3. Results

### 3.1. Child Participants Pharacteristics

Table 2 displays the socio-demographic characteristics of all children who participated in the present study (*n* = 901). The average age of all children was 12.83 ± 0.85 years old. The overall prevalence of overweight or obesity was 28.4% (*n* = 256). Half (50.8%) of the children were female (*n* = 458). The mean value for self-reported weekly pocket money was 21.97 ± 23.96 RMB.

### 3.2. SSB Intake among Children

As shown in Table 3, the mean total amount for weekly consumption of SSBs was 2214.04 ± 2188.62 mL. The most consumed drinks on a weekly basis among children were tea drinks and carbonated drinks.

### 3.3. Relationships between Home-Related Factors, Community Environmental Factors, and SSB Intake

Table 4 reveals that children’s weekly pocket money (OR = 1.696, 95% CI: 1.209–2.380, *p* < 0.01), frequency of purchase SSBs (OR = 1.770, 95% CI: 1.044–3.001, *p* < 0.05), SSB availability at home (OR = 1.503, 95% CI: 1.019–2.216, *p* < 0.05), the number of accessible supermarkets (OR = 1.680, 95% CI: 1.060–2.664, *p* < 0.05), and frequency of weekly visits to convenience stores (OR = 2.088, 95% CI: 1.352–3.224, *p* < 0.01) were all found to be associated with high consumption of sugary drinks among all children.

Among children of normal weight, pocket money (OR = 1.603, 95% CI: 1.079–2.382, *p* < 0.05), SSBs availability at home (OR = 1.593, 95% CI: 1.045–2.427, *p* < 0.05), and the number of accessible supermarkets (OR = 1.797, 95% CI: 1.049–3.079, *p* < 0.05), were associated with higher SSB intake (Table 5). However, among overweight/obesity children, mother’s SSBs intake (OR = 2.081, 95% CI: 1.021–4.239, *p* < 0.05), pocket money (OR = 1.948, 95% CI: 1.004-3.781), frequency of SSB purchase (OR = 3.053, 95% CI: 1.104–8.442, *p* < 0.05), and frequency of weekly visit to convenience stores (OR = 5.273, 95% CI: 2.098–13.256, *p* < 0.01) were associated with higher SSBs intake (Table 6).

Table 7 and Table 8 show the relationships between home-related factors, community environmental factors, and SSBs intake among overweight and obesity children, respectively.

## 4. Discussion

To our knowledge, this is the first study to explore the status of beverage intake and to investigate the relationship between family-related factors and child SSBs consumption among Northeast Chinese children. The present study found there to be high consumption of SSBs among Chinese children. The mean total amount of SSBs consumed on a weekly basis was 2214.04 ± 2188.62 mL. Children’s weekly pocket money, frequency of SSBs purchase, SSBs availability at home, the number of accessible supermarkets, and frequency of weekly visits to convenience stores were associated with a high intake of SSBs among all children. Among children of normal weight, the findings indicated that weekly pocket money, SSBs availability at home, and number of accessible supermarkets were associated with a high SSBs intake. At the same time, frequency of SSBs purchase, mother’s SSB intake, and frequency of weekly visits to convenience stores were associated with a high SSBs intake among children with obesity.

Previous research found that the average urban child in China drank 329 milliliters of soft drinks per day in 1998; in 2008, this number had reached 715 milliliters of soft drinks per day [34]. Moreover, a survey of 53,151 Chinese children aged 6–17 years found that SSBs intake was 2.84 ± 5.26 servings/week (one serving = 250 mL) [16]. In this study, each participant was asked to answer two questions regarding the frequency of consumption of SSBs and the amount of SSBs consumed, of which SSBs included Coca-Cola, Sprite, orange juice, nutritious fast food drinks, and Red Bull. A study of children aged 7 to 18 in Guangdong province showed that teenagers ages 13 to 18 consumed 148.86 ± 8.96 mL sugary drinks every day [15]. To estimate SSBs consumption and frequency, the researchers asked: “How many days and how many servings of SSBs did you have last week?”, where SSBs include carbonated drinks, juices, and sports and sweet tea beverages, and one serving = 250 mL [15]. In the present study, we included 13 different types of SSBs and found there to be an overall high intake of SSBs among Northeast China children.

One finding of the present study was that more pocket money was associated with higher intake of SSBs among children. This finding is consistent with previous research including a cross-sectional study that found that there was a positive dose-response relationship between the amount of pocket money an individual had and BMI, overweight or obesity risk, as well as consumption of snacks, sugar-sweetened beverages, fast food, and so on [35]. These findings filled up the evidence of previous studies from other countries by suggesting that pocket money is a significant risk factor for children’s unhealthy eating behaviors [36] and overweight or obesity [37,38], in addition to other risky behaviors (such as smoking, drug abuse, dental caries) [37,39,40]. Due to lacking data, we cannot explain the specific reasons for this finding. However, the finding suggests the importance of pocket money as a risk factor for many unhealthy lifestyles [37]. Future research should focus on the role of pocket money in children’s lifestyle choices.

The present study also found that home availability of SSBs was significantly associated with SSBs consumption. In this paper, home availability mainly refers to the placement of beverages at home and the frequent purchase of sugary beverages by parents. Among these criteria, the more convenient the placement of beverages at home, the higher the intake of beverages by children. The results of this study are the same as studies conducted in other countries [26,41,42,43]. Thus, home beverages availability is an important factor to consider in future interventions on SSBs consumption.

Several studies conducted in other countries have also observed a link between parents’ and children’s beverages intake [21,27,28,41,42,43]. However, the present study only observed that the mother’s SSBs intake was an important factor; no such relationship was observed between the father’s and the child’s intake. Cultural differences may have contributed to this finding, but nevertheless, these results still highlight the importance of understanding parental role modeling as a potential intervention target for improving children’s consumption of sugary drinks.

Unhealthy environments may play a part in influencing the consumption of sugary drinks [21,44,45]. The present study found that within the community, the number of accessible supermarkets and the frequency of weekly convenience stores visits were strongly correlated with consumption of sugary drinks. In one study, teens were more likely to buy from stores when they lived or went to school in neighborhoods with dense fast-food or convenience stores [46]. Similarly, previous studies have shown that environmental interventions do contribute to obesity interventions [45]. However, this finding is not consistent in the literature. For example, results from the Olympic Regeneration in East London Study [25] showed there was no association between exposure to fast-food restaurants and convenience stores around home, and children’s fast food and beverages intake. This may be due to differences in research design and methodology, as well as cultural, traditional, and dietary norm differences between countries. Even so, it is still important to pay attention to the impact of the community environment and examine the causal association found in the present study in future cohort studies or intervention trials. Ultimately, the government should take relevant measures to reduce the availability of sugar-sweetened beverages in supermarket and convenience stores.

Although the present has many strengths, several limitations should be noted. First, this cross-sectional study does not provide evidence of causal associations or trends in the long term. Second, this study used a self-reported questionnaire to gather information, which poses a high risk of under- or over-reporting. For instance, the variances in findings of SSBs intake may be due to the different definition, measurement, and classification used, primarily because of a limited standardized list of SSBs. There are also some limitations to the definition of variables. For example, the role model is complex, and it was only swapped with parental SSBs consumption in this study. Future study should investigate other factors such as peer effect. Additionally, online purchasing of SSBs was not considered. The other limitation is that the study aimed to determine associations between SSBs consumption and home-related factors and community environmental factors in adolescents of Northeastern China. However, for statistical reasons—that is, the sample size—only the two school students were involved. This may have introduced a bias for comparisons with the general population of China.

## 5. Conclusions

The present study suggests that the weekly consumption of SSBs is high among school-aged Northeastern Chinese children. Our findings highlight the role of home- and community-related factors in SSB consumption. Given the potential negative health effects of high SSBs intake, it is crucial to ensure efforts in reducing SSBs consumption among children. Key points of this study are that beverage availability in home should be monitored, and parental role modeling may influence children’s beverage intake. Therefore, future interventions should focus on limiting the availability and accessibility of sugar-sweetened beverages in home environment and community environment such as educating parents and children to select foods that are part of a healthier dietary pattern and training aimed at the use of children’s pocket money.

## Figures and Tables

**Table 1 nutrients-13-02233-t001:** Screening for overweight and obesity among school-age children and children.

Age (Years Old)	Overweight (kg/m^2^)		Obesity (kg/m^2^)	
	Boy	Girl	Boy	Girl
10	19.6	20.0	22.5	22.1
11	20.3	21.1	23.6	23.3
12	21.0	21.9	24.7	24.5
13	21.9	22.6	25.7	25.6
14	22.6	23.0	26.4	26.3
15	23.1	23.4	26.9	26.9

**Table 2 nutrients-13-02233-t002:** Child participants characteristics (*n* = 901).

Variable	Mean (Median)	SD
Age(years)	12.83	0.85
Sex (*n*, %)		
Male	443(49.2)	
Female	458(50.8)	
Children with Normal weight	645(71.6)	
Children of Overweight and obesity	256(28.4)	
Home-related factors		
Annual household income (*n*, %)		
≤RMB30,000.00	155(17.2)	
RMB30,000.00–100,000.00	408(45.3)	
≥RMB100,000.00	338(37.5)	
Pocket money (RMB ^a^/per week)	21.97(15.00)	23.96
Purchase SSBs ^b^ frequently (*n*, %)		
Yes	723(80.2)	
No	178(19.8)	
SSBs are available at home (*n*, %)		
Yes	573(63.6)	SD
No	328(36.4)	
Parental role model		
Mother SSBs Intake (mL/per week)	239.13(0.00)	370.78
Father SSBs Intake (mL/per week)	213.15(0.00)	334.49
Community environment		
Fast food restaurant		
Number	1.27(1.00)	1.12
Walk time(minute)	13.62(10.00)	10.30
Frequency (times/per week)	0.52(0.00)	0.67
Money (RMB/per visit)	19.15(0.00)	24.90
Supermarket		
Number	1.25(1.00)	0.89
Time(walk)	16.40(15.00)	14.05
Frequency (times/per week)	0.89(1.00)	0.94
Money (RMB/per visit)	57.73(20.00)	77.90
Convenience store		
Number	2.73(2.00)	2.02
Time(walk)	5.73(5.00)	4.18
Frequency (times/per week)	2.17(2.00)	1.78
Money (RMB/per visit)	17.28(10.00)	22.04

^a^: RMB: renminbi; ^b^: SSBs: sweet fruit juice, tea drinks, coffee drinks, instant coffee, sweet milk tea, milk drinks, plant protein drinks, Asian specialty drinks, carbonated drinks, 100% fruit juice drinks, vegetable juices, low-calorie carbonated drinks, energy/sports drinks.

**Table 3 nutrients-13-02233-t003:** Total sugar-sweetened beverage intake in one week (mL).

Types of Beverages	All Children		Normal Weight		Overweight		Obesity		*p* Value
	Mean ± SD	Median	Mean ± SD	Median	Mean ± SD	Median	Mean ± SD	Median	
Water	3996.00 ± 2804.24	3850.00	3936.29 ± 2711.27	3850.00	4169.08 ± 2993.84	3850.00	4129.91 ± 3091.05	3850.00	0.595
Tea drinks	610.88 ± 746.58	500.00	602.88 ± 730.90	500.00	642.59 ± 780.20	500.00	616.07 ± 794.07	500.00	0.853
Carbonated drinks	579.64 ± 865.98	0.00	564.49 ± 842.58	0.00	638.69 ± 928.96	0.00	592.10 ± 918.26	0.00	0.653
Sweetened fruit juice	472.21 ± 757.44	0.00	470.91 ± 750.24	0.00	536.67 ± 816.35	0.00	401.78 ± 723.23	0.00	0.378
Milk beverage	276.82 ± 546.95	0.00	293.55 ± 565.68	0.00	210.14 ± 451.32	0.00	266.05 ± 546.74	0.00	0.263
Energy/sports drinks	221.97 ± 409.98	0.00	219.48 ± 403.74	0.00	244.78 ± 443.93	0.00	208.69 ± 404.70	0.00	0.757
100% Fruit juice	177.72 ± 330.49	0.00	192.25 ± 298.33	0.00	103.60 ± 200.51	0.00	189.38 ± 312.64	0.00	0.004
Low calorie carbonated beverage	120.81 ± 354.58	0.00	126.82 ± 360.45	0.00	126.86 ± 366.28	0.00	80.36 ± 304.27	0.00	0.433
Vegetable juice	96.67 ± 265.91	0.00	95.89 ± 261.44	0.00	72.18 ± 274.23	0.00	122.12 ± 279.58	0.00	0.341
Vegetable protein beverage	120.14 ± 250.58	0.00	130.53 ± 260.86	0.00	68.32 ± 186.24	0.00	125.40 ± 254.50	0.00	0.030
Coffee drinks	122.99 ± 275.93	0.00	136.98 ± 292.14	0.00	97.81 ± 251.48	0.00	73.76 ± 189.05	0.00	0.044
Asian specialty beverage ^a^	70.85 ± 240.49	0.00	66.40 ± 225.21	0.00	63.65 ± 241.86	0.00	103.65 ± 308.99	0.00	0.296
Sweet milk tea	56.33 ± 97.65	0.00	59.77 ± 99.30	0.00	47.41 ±95.41	0.00	48.27 ± 90.68	0.00	0.261
Instant coffee	5.55 ± 14.18	0.00	6.20 ± 14.76	0.00	3.61 ± 12.67	0.00	4.32 ± 12.37	0.00	0.095
Total amount	2214.04 ± 2188.62	1580.00	2233.84 ± 2203.82	1590.00	2186.03 ± 2140.79	1550.00	2141.93 ± 2190.47	1528.00	0.928

^a^: Asian specialty beverage: a beverage peculiar to Asia, such as fruit-fermented beverages.

**Table 4 nutrients-13-02233-t004:** Factors associated with sugar-sweetened beverages intake in participants.

Variable	Model 1 ^a^	Model 2 ^b^
Home-related factors		
Income (RMB)		
30,000.00–100,000.00	0.901(0.561,1.447)	0.812(0.474,1.390)
≥100,000.00	1.068(0.654,1.744)	1.043(0.599,1.818)
Pocket money (>20.00 RMB/per week)	1.696(1.209,2.380) **	1.346(0.919,1.970)
Purchase SSBs frequently	1.788(1.111,2.877) *	1.770(1.044,3.001) *
SSBs are available at home	1.512(1.062,2.154) *	1.503(1.019,2.216) *
Parental role model		
Mother SSBs Intake (per week)	1.352(0.937,1.952)	1.403(0.932,2.112)
Father SSBs Intake (per week)	1.379(0.955,1.992)	1.103(0.734,1.657)
Community environmental factors		
Fast food restaurant		
Number (>1)		0.733(0.466,1.152)
Walk time (>10 min)		1.200(0.808,1.782)
Frequency (>1 times/per week)		1.625(0.768,3.439)
Money (>21.00 RMB/per visit)		0.821(0.338,1.741)
Supermarket		
Number (>1)		1.680(1.060,2.664) *****
Walk time (>15 min)		1.311(0.863,1.991)
Frequency (>1 times/per week)		1.362(0.729,2.545)
Money (>30.00 RMB/per visit)		0.840(0.566,1.246)
Convenience store		
Number (>2)		1.100(0.738,1.640)
Walk time (>5 min)		1.226(0.807,1.864)
Frequency (>2 times/per week)		2.088(1.352,3.224) **
Money (>10.00 RMB/per visit)		1.308(0.820,2.087)

^a^: Model 1: home-related factors—Income, Pocket money, Purchase SSBs frequently, SSBs are available at home, Parental role model, Mother SSBs intake, Father SSBs intake; Hosmer–Lemeshow test: *p* = 0.823 > 0.05; the quality of the model is good. ^b^: Model 1: home-related factors—Income, Pocket money, Purchase SSBs frequently, SSBs are available at home, Parental role model, Mother SSBs intake, Father SSBs intake, and Community environmental factors—Fast food restaurant, Supermarket, Convenience store; Hosmer–Lemeshow test: *p* = 0.216 > 0.05; the quality of the model is good. Adjusted by age, sex; * *p* < 0.05, ** *p* < 0.01.

**Table 5 nutrients-13-02233-t005:** Factors associated with sugar-sweetened beverage intake in normal weight children.

Variable	Model 1 ^a^	Model 2 ^b^
Home-related factors		
Income (RMB *)		
10,000.00–100,000.00	1.044(0.602,1.813)	1.049(0.566,1.946)
≥100,000.00	1.042(0.590.1.840)	1.027(0.542,1.946)
Pocket money (>20.00 RMB/per week)	1.603(1.079,2.382) *	1.312(0.837,2.056)
Purchase SSBs frequently	1.740(0.966,3.134)	1.593(0.977,2.437)
SSBs are available at home	1.593(1.045,2.427) *	1.543(0.977,2.437)
Parental role model		
Mother SSBs Intake (per week)	1.265(0.825,1.941)	1.279(0.803,2.039)
Father SSBs Intake (per week)	1.411(0.916,2.174)	1.151(0.718,1.847)
Community environmental factors		
Fast food restaurant		
Number (>1)		0.711(0.420,1.203)
Walk time (>10 min)		1.174(0.737,1.869)
Frequency (>1 times/per week)		1.452(0.560,3.762)
Money (>21.00 RMB/per visit)		1.012(0.388,2.642)
Supermarket		
Number (>1)		1.797(1.049,3.079) *
Walk time (>15 min)		1.416(0.876,2.287)
Frequency (>1 times/per week)		1.452(0.560,3.762)
Money (>30.00 RMB/per visit)		0.822(0.518,1.303)
Convenience store		
Number (>2)		1.066(0.665,1.711)
Walk time (>5 min/per week)		1.290(0.780,2.132)
Frequency (>2 times)		1.562(0.932,2.617)
Money (>10.00 RMB/per visit)		1.022(0.594,1.756)

^a^: Model 1: home-related factors—Income, Pocket money, Purchase SSBs frequently, SSBs are available at home, Parental role model, Mother SSBs intake, Father SSBs intake; Hosmer–Lemeshow test: *p* = 0.772 > 0.05; the quality of the model is good. ^b^: Model 1: home-related factors—Income, Pocket money, Purchase SSBs frequently, SSBs are available at home, Parental role model, Mother SSBs intake, Father SSBs intake, and Community environmental factors—Fast food restaurant, Supermarket, Convenience store; Hosmer–Lemeshow test: *p* = 0.809 > 0.05; the quality of the model is good. Adjusted by age, sex; * *p* < 0.05.

**Table 6 nutrients-13-02233-t006:** Factors associated with beverages intake in overweight/obesity children.

Variable	Model 1 ^a^	Model 2 ^b^
Home-related factors		
Income (RMB)		
30,000.00–100,000.00	0.650(0.253,1.666)	0.468(0.138,1.581)
≥100,000.00	0.975(0.372,2.555)	1.205(0.345,4.209)
Pocket money (>20.00 RMB/per week)	1.948(1.004,3.781) *	1.529(0.667,3.510)
Purchase SSBs frequently	1.981(0.876,4.476)	3.053(1.104,8.442) *
SSBs are available at home	1.342(0.698,2.578)	1.599(0.698,3.666)
Parental role model		
Mother SSBs Intake (per week)	2.081(1.021,4.239) *	2.314 (0.892,6.004)
Father SSBs Intake (per week)	1.208(0.595,2.450)	0846(0.345,2.079)
Community environmental factors		
Fast food restaurant		
Number (>1)		0.795(0.286,2.208)
Walk time (>10 min)		0.890(0.385,2.058)
Frequency (>1 times/per week)		2.408(0.595,9.747)
Money (>21.00 RMB/per visit)		0.644(0.165,2.507)
Supermarket		
Number (>1)		1.642(0.623,4.325)
Walk time (>15 min)		0.803(0.313,2.060)
Frequency (>1 times/per week)		1.027(0.325,3.247)
Money (>30.00 RMB/per visit)		1.265(0.520,3.079)
Convenience store		
Number (>2)		0.907(0.392,2.099)
Walk time (>5 min)		0.902(0.394,2.065)
Frequency (>2 times/per week)		5.273(2.098,13.256) **
Money (>10.00 RMB/per visit)		2.765(0.967,7.909)

^a^: Model 1: home-related factors—Income, Pocket money, Purchase SSBs frequently, SSBs are available at home, Parental role model, Mother SSBs intake, Father SSBs intake; Hosmer–Lemeshow test: *p* = 0.811 > 0.05; the quality of the model is good. ^b^: Model 1: home-related factors—Income, Pocket money, Purchase SSBs frequently, SSBs are available at home, Parental role model, Mother SSBs intake, Father SSBs intake, and Community environmental factors—Fast food restaurant, Supermarket, Convenience store; Hosmer–Lemeshow test: *p* = 0.981 > 0.05; the quality of the model is good. Adjusted by age, sex; * *p* < 0.05, ** *p* < 0.01.

**Table 7 nutrients-13-02233-t007:** Factors associated with beverages intake in overweight children.

Variable	Model 1 ^a^	Model 2 ^b^
Home-related factors		
Income (RMB)		
30,000.00–100,000.00	0.500(0.142,1.716)	0.865(0.145,5.145)
≥100,000.00	1.130(0.309,4.131)	12.233(1.529,97.862) *
Pocket money (>20.00 RMB/per week)	1.743(0.724,4.196)	0.920(0.281,3.010))
Purchase SSBs frequently	2.508(0.816,7.708)	12.392(2.282,67.279) **
SSBs are available at home	1.595(0.664,3.833)	4.697(1.278,17.275) *
Parental role model		
Mother SSBs Intake (per week)	1.621(0.666,3.942)	1.918(0.524,7.024)
Father SSBs Intake (per week)	1.928(0.792,4.696)	1.807(0.551,5.931)
Community environmental factors		
Fast food restaurant		
Number (>1)		2.667(0.622,11.430)
Walk time (>10 min)		0.464(0.141,1.527)
Frequency (>1 times/per week)		0.309(0.048,1.968)
Money (>21.00 RMB/per visit)		1.602(0.261,9.814)
Supermarket		
Number (>1)		2.082(0.526,8.236)
Walk time (>15 min)		5.939(1.286,27.424)
Frequency (>1 times/per week)		2.226(0.464,10.688)
Money (>30.00 RMB/per visit)		0.923(0.263,3.239)
Convenience store		
Number (>2)		0.246(0.065,0.935)
Walk time (>5 min)		1.141(0.369,3.525)
Frequency (>2 times/per week)		4.933(1.390,17.514) *
Money (>10.00 RMB/per visit)		3.936(0.825,18.775)

^a^: Model 1: home-related factors—Income, Pocket money, Purchase SSBs frequently, SSBs are available at home, Parental role model, Mother SSBs intake, Father SSBs intake; Hosmer–Lemeshow test: *p* = 0.160 > 0.05; the quality of the model is good. ^b^: Model 1: home-related factors—Income, Pocket money, Purchase SSBs frequently, SSBs are available at home, Parental role model, Mother SSBs intake, Father SSBs intake, and Community environmental factors—Fast food restaurant, Supermarket, Convenience store; Hosmer–Lemeshow test: *p* = 0.721 > 0.05; the quality of the model is good. Adjusted by age, sex; * *p* < 0.05, ** *p* < 0.01.

**Table 8 nutrients-13-02233-t008:** Factors associated with beverages intake in obesity children.

Variable	Model 1 ^a^	Model 2 ^b^
Home-related factors		
Income (RMB)		
30000.00–100000.00	1.202(0.294,4.916)	1.197(0.131,10.920)
≥100000.00	1.197(0.293,4.885)	0.768(0.082,7.208)
Pocket money (>20.00 RMB/per week)	2.223(0.830,5.956)	5.368(1.056,27.286) *
Purchase SSBs frequently	1.782(0.515,6.162)	1.570(0.259,9.529)
SSBs are available at home	1.018(0.391,2.650)	0.656(0.160,2.689)
Parental role model		
Mother SSBs Intake (per week)	2.537(0.807,7.979)	3.829(0.708,20.699)
Father SSBs Intake (per week)	0.647(0.202,2.067)	0.361(0.062,2.112)
Community environmental factors		
Fast food restaurant		
Number (>1)		0.363(0.111,1.184)
Walk time (>10 min)		3.257(0.743,14.279)
Frequency (>1 times/per week)		2.408(0.595,9.747)
Money (>21.00 RMB/per visit)		2.216(0.522,8.659)
Supermarket		
Number (>1)		3.163(0.385,26.014)
Walk time (>15 min)		0.190(0.033,1.094)
Frequency (>1 times/per week)		1.261(0.142,11.217)
Money (>30.00 RMB/per visit)		4.896(0.810,29.592)
Convenience store		
Number (>2)		5.413(1.179,24.843) *
Walk time (>5 min)		0.558(0.118,2.636)
Frequency (>2 times/per week)		8.075(2.361,27.617) **
Money (>10.00 RMB/per visit)		1.342(0.287,6.272)

^a^: Model 1: home-related factors—Income, Pocket money, Purchase SSBs frequently, SSBs are available at home, Parental role model, Mother SSBs intake, Father SSBs intake; Hosmer–Lemeshow test: *p* = 0.203 > 0.05; the quality of the model is good. ^b^: Model 1: home-related factors—Income, Pocket money, Purchase SSBs frequently, SSBs are available at home, Parental role model, Mother SSBs intake, Father SSBs intake, and Community environmental factors—Fast food restaurant, Supermarket, Convenience store; Hosmer–Lemeshow test: *p* = 0.952 > 0.05; the quality of the model is good. Adjusted by age, sex; * *p* < 0.05, ** *p* < 0.01.

## Data Availability

The data presented in this study are available on request from the corresponding author. The data are not publicly available due to privacy and ethical restrictions.

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
