# Peer review of "Sugar-Sweetened Beverages Consumption and Associated Factors among Northeastern Chinese Children"

_nutrients, 2021, doi:10.3390/nu13072233_

Round 1

Reviewer 1 Report

The cross-sectional design needs more explanation about the refusals to answer the questionnaire specifying the initial and final sample

The limitation section should be improved and expanded

Gross results should be specified separating overweight and obesity

Table 1 is surprising the large number of obese children, this must be ensured that it is well defined

Reviewer 2 Report

Article:

 Sugar-Sweetened Beverage Consumption and Associated Fac-2 tors among Northeastern Chinese Children. By Xuxiu Zhuang, Yang liu, Joel Gittelsoh, Emma Lewis, Shenzhi Song, Yanan Ma and Deliang Wen

The article investigates sugar-sweetened beverage consumption in Chinese children using a cross-sectional design. While the general research question is of interest, however, the article lacks behind in several aspects. Firstly, the article is poorly written and riddled with spelling and grammatical errors (i.e. abstract, but also within the main document).

Comments regarding methods:

Recruitment and selection of participants are not well described and reads like that there was no informed consent procedure in place with potential participants having had the freedom to take part or not. In addition, readdressing participants to fill in missing data is a very unusual approach to research.

Questionnaire for beverage consumption reports are based on images of number of bottles consumed – this might be possible for readily bottled fluids, but other drinks are not bottled and therefore it seems that this led to underreporting of fluid intake. In the mean, only a total intake of about 6 to 7 litres per week is reported, while a much higher total intake would be expected; Zhang et al. 2018 reported about 10 litres per week for rural China, while real consumption may be rather around 12 litres. Design and way of data collection might have resulted to underreporting because of obvious social desirability of low consumption.  

Screening of buying activity of SSB was only addressing shops – not any deliveries towards home. Is this not relevant for the population tested?

Several measures are not consistent in reporting and problematic in labelling. Soft drink consumption of parents is measured, according to methods, only on the weekends as a measure of parental role model, while in tables this is expressed per week. In addition, the idea or role model is more complex and should not simply be swapped with a measure of consumption. In addition, parents, as a valuable role model, are heavily in decline in the age group investigated. Authors should have rather investigated friends’ consumption instead.

Results:

Numeric values should be given only with significant digits, too often the given values assume a measuring accuracy that is not available. Within tables, errors are apparent in the labelling unit and dimensions. Reporting of statistics is incomplete, logistic regression only reports odds ratios, the quality of the models is not reported at all.

Discussion is problematic because it heavily focuses on parameters that are only slightly higher than others in odds ratio in the logistic regression; in addition, because of the incomplete report of the model properties, interpretations are exaggerated. Findings are not surprising due to the selection of measures which are serving a more self-fulfilling prophecy that if children have more many to spend with higher opportunity to buy SSB, it will happen.

In consequence, I recommend a rewrite.

Reviewer 3 Report

This is a very interesting article that includes a very actual topic of great concern at the international level, such as SSB consumption. I find it really important to evaluate the factors that increase their consumption in order to find ways to generate policies that form consumers to prepare and select foods that are part of a healthier dietary pattern.  Following you will find a few comments regarding the manuscript:

ABSTRACT

Line 26: I see the phrase "highlight the purpose of the study" at the end of the abstract, I think the phrase is included there by mistake.

INTRODUCTION

Line 31: should say "problems"

Line 36-37: In my opinion over-comsumption of almost every foods can lead to overweight and/or obesity, research should also focus on dietary patterns and not only individual foods.  For sure SBBs are not part of a healthy dietary pattern, but not responsible of overweight and obesity by themselves.

MATERIALS AND METHODS

Line 71: "Shenyang City" need initial uppercase letter; and, if participants were "ramdomly selected" the randomization process should be explained.

Line 99: Correct the closing quotation mark in "How much per week"

Line 104:  Change "estimates" for "estimate"

Line 105: Please add a space between "1" and "to"

Table 3:

-Please add (ml), at the end of the table title. 

-I would like to see p-values from comparison between Normal weight and Oberweiht/obese children, and discusion/conclusion regarding results.

-As it seems distributions are not normal, so comparisons should be using median values. This table should include in footnotes what is "Asian specialty beverage" made from.

Table 4-6: Please explain briefly and in a footnote model 1 and 2, I understand you explained it in the methodology, but it is ideal that each table can be interpreted on its own.

Line 229: I think there is some word/s missing after the word  "limited" at the end of this line.

Conclusions:

¿What about interventions that include educating parents and children to select foods that are part of a healthier dietary pattern and that there is good availability of those foods in communities; or training aimed at the use of children's pocket money, etc.

Round 2

Reviewer 2 Report

Authors revised the manuscript and addressed the comments which have been raised.

This manuscript is a resubmission of an earlier submission. The following is a list of the peer review reports and author responses from that submission.